# Satellite-Based Optimization and Planning of Urban Ventilation Corridors for a Healthy Microclimate Environment

**Deming Gong, Xiaoyan Dai * and Liguo Zhou ***

Department of Environmental Science and Engineering, Fudan University, Shanghai 200438, China; 20210740107@fudan.edu.cn
* Correspondence: xiaoyandai@fudan.edu.cn (X.D.); lgzhou@fudan.edu.cn (L.Z.)

**Abstract:** Urban ventilation corridors (UVCs) have the potential to effectively mitigate urban heat islands and air pollution. Shanghai, a densely populated city located in eastern China, is among the hottest cities in the country and requires urgent measures in order to enhance its ventilation system. This study introduces a novel approach that integrates land surface temperature retrieval, $PM_{2.5}$ concentration retrieval, and wind field simulation to design UVCs at the city level. Through remote sensing data inversion of land surface temperature (LST) and $PM_{2.5}$ concentration, the study identifies the action spaces and compensation spaces for UVCs. The Weather Research and Forecasting (WRF) model, coupled with the multilayer urban scheme Building Effect Parameterization (BEP) model, is employed to numerically simulate and analyze the wind field. Based on the identification of thirty high-temperature zones and high $PM_{2.5}$ concentration zones as action spaces, and twenty-two low-temperature zones and low $PM_{2.5}$ concentration zones as compensation spaces in Shanghai, the study constructs seven first-class ventilation corridors and nine secondary ventilation corridors according to local circulation patterns. Unlike previous UVC research, this study assesses the cleanliness of cold air, which is a common oversight in UVC planning. Ignoring the assessment of cold air cleanliness can result in less effective UVCs in improving urban air quality and even exacerbate air pollution in the central city. Therefore, this study serves as a crucial contribution by rectifying this significant deficiency. It not only provides a fresh perspective and methodology for urban-scale ventilation corridor planning but also contributes to enhancing the urban microclimate by mitigating the effects of urban heat islands and reducing air pollution, ultimately creating a livable and comfortable environment for urban residents.

**Keywords:** ventilation corridor; land surface temperature (LST); $PM_{2.5}$ concentration; wind environment; Shanghai

## 1. Introductions

At present, China is in a stage of rapid urbanization. This urbanization, which is characterized by population growth, land expansion, and changes in surface traits, has had a far-reaching impact on the local climate. The urban heat island (UHI) effect, air pollution, and a weaker air exchange have become important issues that need to be solved for the sake of sustainable urban development [1–5]. In recent years, many researchers and policy-makers have continuously explored ways in which to mitigate the deterioration of the urban microclimate environment from all aspects of urban planning, construction, operation, and maintenance [6–9]. Among these aspects, the study of urban ventilation corridors (UVCs) has been a research hotspot in recent years [10–15]. The global outbreak of the novel coronavirus pneumonia epidemic has once again indicated the importance of improving the urban ventilation capacity by optimizing UVCs [16].

UVCs are passages through which fresh air from the suburbs can enter an urban area. These corridors improve the microcirculation of an urban system, mitigate the heat island effect, increase the thermal comfort of urban residents, and advance the livability

of the city [17,18]. At present, these corridors have become a hot topic of theoretical discussions and in practice in land and spatial planning and healthy city construction [19]. Chinese governments at all levels have carried out relevant work to actively promote the reform and implementation of territorial and spatial planning and promulgated a series of relevant policy documents, such as the "Several Opinions of the Central Committee of the Communist Party of China and the State Council on Establishing and Supervising the Implementation of a Territorial Planning System" [20] and the "Notice on Comprehensively Carrying out Territorial and Spatial Planning" [21] documents, both of which propose urban ventilation corridor planning. In recent years, most provinces and cities in China have carried out the construction of ventilation corridor projects. For example, Du et al. [22] proposed the construction of ventilation corridors in the central urban area of Beijing based on meteorology and the GIS technique, which was applied to the overall plan of Beijing. Ren et al. [23] built the UVCs in Chengdu and incorporated them into the urban ecological corridors. Xie et al. [24] proposed a new approach for ventilation corridor recognition based on circuit theory, taking Wuhan as the research area. Based on the method of least cost path (LCP), Fang et al. [25] analyzed the ventilation corridor in the core urban area of Hefei, Anhui Province. Lai et al. [26] proposed a sky-view factor (SVF)-based method for the potential wind corridor detection in a built-up urban area of Zhangzhou, Fujian Province. Liu et al. [27] devised an integrated air ventilation assessment (IAVA) method by which to assess the urban wind environment using multi-source data, with Shenzhen as an example.

At present, wind environment assessment methods in urban planning mainly include numerical simulations, wind tunnel tests, and field tests [28]. However, each wind environment assessment method has certain limitations [29]. The spatial resolution of mesoscale meteorological simulation results cannot reflect the fine impact of the overall layout of buildings on the urban internal wind field on the overall planning scale of the city [30]. Computational fluid dynamics (CFD) simulation can show the refined wind field inside the block, but its simulation range is small, and the high-resolution wind field simulation at the whole urban scale cannot be completed quickly due to the large computation requirement [31,32]. The long evaluation cycle, high cost, and low universality of wind tunnel experiments make them unsuitable for urban-scale ventilation evaluations [33,34]. The test ranges of field experiments are relatively small, generally limited to urban streets or local areas, and futher limited by problems such as few data points; thus, they do not have a high applicability [35]. In recent years, the combination of geographic information system (GIS) and remote sensing (RS) techniques with meteorology has provided new means for urban-scale ventilation environment assessment, integrating architectural information such as the building height and density in urban areas. For example, the GIS-based method identifies areas with high ventilation potential by analyzing the roughness length of the underlying surface [36,37]. The frontal area index (FAI) is used by scholars to identify and analyze the UVC [38]. Based on multi-source data, the potential ventilation corridors are created using the least cost path (LCP) method [18,25,27]. The above research studies on UVCs primarily employed RS techniques, GIS techniques, and CFD simulations to analyze the spatial compositions and action mechanisms of UVCs. However, they failed to consider the influence of meso-scale atmospheric circulation, spatiotemporal characteristics, and weather-related factors. Additionally, they overlooked the assessment of air quality in cold air areas. According to a study by Han et al. [39], polluted cold air can be drawn into the central city area through ventilation corridors if the urban compensation space is located in a region with high pollutant concentrations and is influenced by prevailing winds. Therefore, it is crucial for cities to assess the air quality of compensation spaces when designing urban ventilation corridors.

This study proposes a comprehensive method for urban-scale ventilation corridor optimization and planning that combines GIS and RS techniques and wind environment numerical simulations. This method uses statistical analysis, numerical simulation, and retrieval of land surface temperatures (LSTs) and $PM_{2.5}$ concentrations to build an urban-scale

ventilation corridor system with meteorological observation data and satellite remote sensing data. In contrast to the previous design methods of urban-scale ventilation corridors, the spatial distribution of the urban thermal field is combined with the $PM_{2.5}$ concentration field in this study to determine the action and compensation spaces of UVCs and the source and destination of the compensation air mass. On the one hand, the airflow trajectory and its origins can be more accurately positioned, which helps to improve the effect of the ventilation corridor; on the other hand, when determining the compensation space, the air pollution of the compensatory space is considered, which ensures the cleanliness of the air in the compensatory space to avoid aggravation of the air pollution in the central urban area and deterioration of the urban microclimate environment in ventilation corridor planning.

According to the above methods, Shanghai is used as the study area in this paper to conduct research on urban ventilation corridor planning. On the one hand, this study can provide ideas and a methodological basis for urban atmospheric environmental management and help further refine the planning and construction of ventilation corridors. On the other hand, through analyses of the ventilation corridor planning cases in Shanghai, the existing planning methods for UVCs and for improving the urban microclimate environment can be enhanced from an urban planning perspective.

## 2. Description of Study Area and Data Used

### 2.1. Study Area

Shanghai is located from 120°51′ to 122°12′ E and 30°40′ to 31°53′ N. It is located on the south bank of the Yangtze River Estuary in the middle of the mainland China coastline (Figure 1). It is bordered by the East China Sea to the east, Hangzhou Bay to the south, and the mouth of the Yangtze River to the north. It is approximately 100 km wide from east to west and 130 km long from north to south, with a total area of approximately 8239 km². The average altitude of Shanghai is approximately 4 m, and the terrain is flat with a small relative height difference.

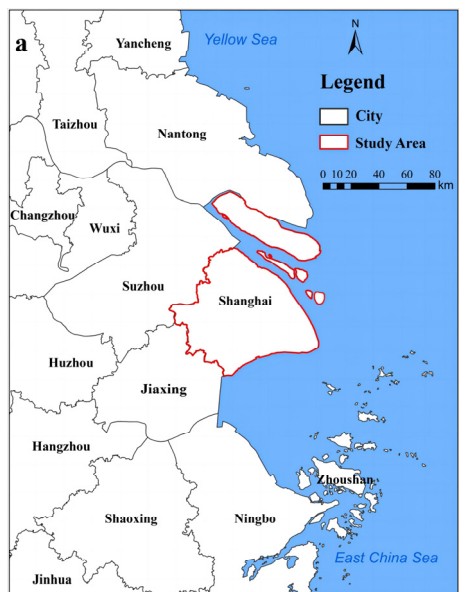
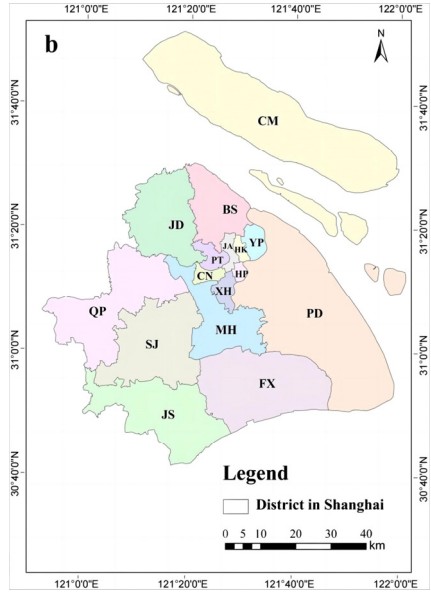

**Figure 1.** Map of the study area. (**a**) Location of Shanghai in the Yangtze River Delta. (**b**) Districts in Shanghai. Note: CM means Chongming; JD means Jiading; BS means Baoshan; PT means Putuo; JA means Jing'an; HK means Hongkou; YP means Yangpu; CN means Changning; HP means Huangpu; XH means Xuhui; QP means Qingpu; SJ means Songjiang; MH means Minhang; PD means Pudong new area; JS means Jinshan; FX means Fengxian.

With the rapid economic and urban development taking place in Shanghai, the impact of human activities on the local climate and atmospheric environment has become increas-

ingly significant. The urban heat island effect in Shanghai is spreading, and its affected areas are gradually expanding (Figure 2a). The heat island region has seen a gradual increase in size from 2481 km$^2$ in 2002 to 3250 km$^2$ in 2020. Moreover, A study of 11 years of weather observation records measured at a height of 10 m above the ground reveals that the wind speed in Shanghai has been showing an annual decreasing trend from 2011 to 2021 (Figure 2b). These observations indicate a decline in both the thermal and wind environments. Therefore, UVCs should be constructed in Shanghai to improve the urban air circulation system by coordinating its climatic and environmental characteristics.

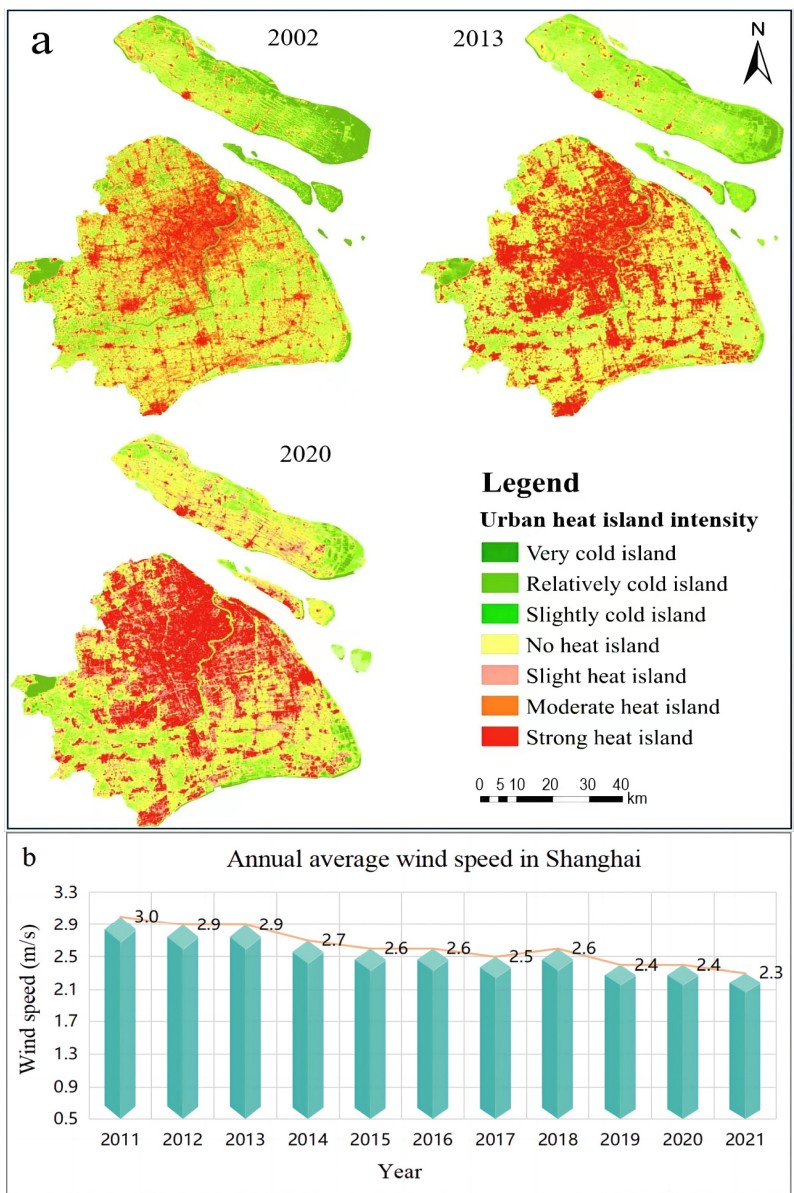

**Figure 2.** (**a**) Spatial distribution of UHIs in Shanghai in different years. (**b**) Annual average wind speed variation from 2011 to 2021.

*2.2. Data Sources*

Data were obtained from various sources, as indicated in Table 1. The data used to retrieve LST were specifically selected for the summer period, when the UHI effect in Shanghai is particularly pronounced. As June and July are characterized by rainy seasons, there is higher land cloud cover during this time, which is unsuitable for LST inversion. Consequently, this study opted for the month of August, when high summer

temperatures exert a greater influence. On the other hand, the data chosen to retrieve $PM_{2.5}$ concentration were focused on the winter season, during which the air quality in Shanghai tends to be worse compared to other seasons. Similar to the LST retrieval, February was selected due to its land cloud cover characteristics. For the statistical analysis of the background wind environment, hourly weather station data from the meteorological observation stations in Shanghai covering the period from 2011 to 2021 were utilized. Additionally, the NASADEM data and GlobeLand30 data were employed as the elevation and land cover data, respectively.

**Table 1.** Data Sources.

| Data | Spatial Resolution | Data Source | Website Link | Acquisition Time |
|---|---|---|---|---|
| Landsat 8 | 30/100 m | The official website of the United States Geological Survey (USGS) | https://earthexplorer.usgs.gov/ (accessed on 18 March 2022) | 10:25 Beijing time on 16 August 2020 |
| The Chinese Gaofen-1 satellite WFV4 | 16 m | The official website of the China Resources Satellite Application Center | https://www.cresda.com/ (accessed on 18 March 2022) | 10:52 Beijing time on 21 February 2021 |
| NASADEM | 30 m | The National Aeronautics and Space Administration (NASA) | https://earthdata.nasa.gov/esds/competitive-programs/measures/nasadem (accessed on 18 March 2022) | 2020 |
| GlobeLand30 | 30 m | The official website of the GlobeLand30 | http://globallandcover.com/ (accessed on 18 March 2022) | 2020 |
| Hourly $PM_{2.5}$ monitoring data | | The Shanghai state-controlled $PM_{2.5}$ monitoring sites | https://air.cnemc.cn:18007/ (accessed on 18 March 2022) | 11:00 Beijing time on 21 February 2021 |
| Hourly meteorological data | | The meteorological observation stations in Shanghai | http://data.cma.cn/ (accessed on 18 March 2022) | January 2011– December 2021 |

## 3. Methodology

### 3.1. Theory of Urban Ventilation Corridor Construction

Local circulation is mainly forced by the heterogeneity of the power and heat of the land surface. It is an important part of the mesoscale circulation, which has obvious or even dominant effects on the weather, both locally and even on a large scale [40]. The German researcher, Kress [41], first proposed an evaluation standard for the climate function of land surface according to local circulation patterns and classified the land surface into action spaces (wirkungsraum), compensation spaces (ausgleichsraum), and air guidance channels (luftleitbahn).

#### 3.1.1. Compensation Space

From the German theory of urban ventilation corridor construction, it can be seen that compensation spaces are areas with a relatively weak heat island effect and relatively light air pollution in the suburbs or within a city. They are ecological cold sources that can produce fresh and cold air, and they are usually the starting points in the design of ventilation corridors. According to the land use types and ecoclimatic functional characteristics of planned land for ventilation corridors, Liu and Shen [42] subdivided the compensation space into cold air-generating areas, large urban green spaces, and suburban woodlands.

Cold air-generation areas usually produce cold air at night, but the cooling degree of near-surface air layers at night is related to soil properties and land surface types. The undeveloped areas with small surface heat capacity and thermal conductivity are ideal cold air-generation areas, of which cultivated land and grasslands are the most ideal, followed

by hillside woodlands [43]. In calm weather, the local airflow formed between the cold air-generation area and the built-up area is very important to promote urban ventilation [44]. Therefore, urban planning should focus on protecting grasslands and cultivated land in the suburbs as an important ventilation entrance to UVCs.

The suburban woodlands not only have thermal compensation function, but also improve the air quality in the surrounding area [45]. Therefore, in urban construction, it is necessary to maintain and develop the thermal compensation function of suburban woodlands and maximize the use of UVCs to introduce fresh cold air from the suburban woodlands to the urban area so as to alleviate the UHI effect and improve the air quality of the urban area.

The climate regulation efficiency of green space depends on many factors, such as scale, vegetation structure, and aerodynamic roughness, among which the "scale" is the determining factor [46,47]. Yu et al. [48] applied spatial analysis technique to comprehensively evaluate the distribution of green space and the heat island effect in central urban areas. They found that the larger and more concentrated the distribution of green space, the better the effect of mitigating the UHI effect. Obviously, the construction of large green space in urban areas and the development of a perfect greening network are important measures to build an urban ventilation system.

### 3.1.2. Action Space

Action space refers to areas that need to improve the wind environment or reduce pollution. In order to make full use of the effect of urban ventilation system built by local circulation, the construction of action space should follow the following principles. First, surface materials with less heat capacity should be selected in the process of urban construction to alleviate thermal pollution, such as reducing the construction of nonpermeable ground in parking lots and road facilities, reducing the building density of urban areas, and improving greening rates [49]. Second, pollutant emissions from urban pollution sources should be controlled to alleviate air pollution problems, for example, by avoiding arranging the pollution sources at the ventilation entrances of dominant wind direction, cold air-generation areas, and UVCs, reducing the use of private motor vehicles, encouraging public transportation, and vigorously promoting the development and application of clean energy. Third, the impact range of the compensatory air mass in the action space should be expanded as much as possible, and the penetration of cold air in calm weather should be improved to promote the climate regulation function of the compensatory air mass, for example, by controlling the height, density, and arrangement of the building complex in the action space and building a loose open complex with a high greening rate on both sides of the ventilation corridor of the action space so as to avoid cold air heating up early and causing convection before reaching the city center.

### 3.1.3. Ventilation Corridor

Ventilation corridors, also known as air guide channels, are connecting channels that guide air from the compensation space to the action space. Based on many studies on ventilation corridors [23,41,42], the following construction strategies are summarized: (1) the aerodynamic roughness of the ventilation corridor ($Z_0$) should be ≤0.5 m; (2) the length of the ventilation corridor should be more than 1000 m in one direction, the width should be more than 30 m, and the width of the first-level ventilation corridor should be at least 200 m; (3) the width of any obstacle perpendicular to the direction of airflow in the ventilation corridor should be less than 10% of the total width of the channel, the height of any obstacle should be limited to less than 10 m, and the ratio of the height of any two adjacent obstacles to the horizontal spacing should not exceed 0.1.

### 3.2. Design and Planning Methods of UVCs

This study first retrieves the LSTs and $PM_{2.5}$ concentrations of Shanghai from remote sensing data and analyzes the spatial differences in the thermal characteristics of the land

surface and PM$_{2.5}$ concentration. On this basis, the action space and compensation space are determined according to local circulation patterns, and the driving mechanism of urban ventilation is revealed. Then, the Weather Research and Forecasting (WRF) [50] model, coupled with the multilayer urban scheme Building Effect Parameterization (BEP) [51], is used to simulate the wind field in Shanghai. Finally, the potential ventilation corridors in Shanghai are constructed according to the distribution of the action and compensation spaces, the dominant wind directions in winter and summer, and the wind field simulation results. Figure 3 shows the research workflow.

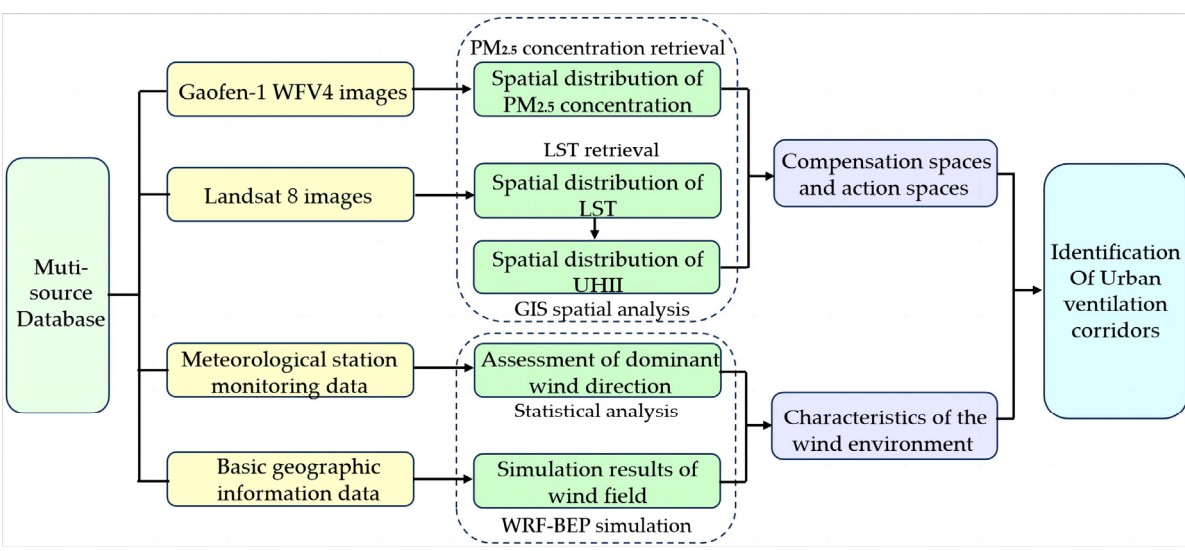

**Figure 3.** Research workflow chart.

### 3.2.1. Analysis of the Thermal Environment

The LSTs are retrieved from remote sensing data mainly using the radiation transmission equation method, mono-window algorithm, and single-channel algorithm [52]. Because Shanghai has a subtropical monsoon climate, it is hot and humid and has high atmospheric water vapor content in summer. When the atmospheric moisture content is high, the single-window algorithm and the single-channel algorithm result in large LST estimation errors [53,54]. Therefore, the radiation transmission equation method is used for LST retrieval in this paper. The specific retrieval process can be found in the literature [55].

According to the LST retrieval results, the difference between the LST of the inner urban area and the suburbs is defined as Urban Heat Island Intensity (UHII) [56,57]. In this paper, the difference between the urban surface temperature and the average LST of the cold surface (vegetation cover and water) is used to calculate the UHII [55]. Furthermore, the UHII is divided into seven levels (Table 2): very cold island, relatively cold island, slightly cold island, no heat island, slight heat island, moderate heat island, and strong heat island [23].

**Table 2.** Classification of UHII.

| Classification | Description | Daily UHII (°C) |
|:---:|:---:|:---:|
| 1 | Very cool island | $\leq -7.0$ |
| 2 | Relatively cool island | $-7.0 \sim -5.0$ |
| 3 | Slightly cool island | $-5.0 \sim -3.0$ |
| 4 | No heat island | $-3.0 \sim 3.0$ |
| 5 | Slightly heat island | $3.0 \sim 5.0$ |
| 6 | Relatively heat island | $5.0 \sim 7.0$ |
| 7 | Strong heat island | $\geq 7.0$ |

The detailed calculation is as follows:

$$\text{UHII}_i = T_i - \frac{1}{n}\sum\nolimits_{j=1}^{n} T_{coolj},\tag{1}$$

where $\text{UHII}_i$ is the intensity of the surface heat island corresponding to the *i*th pixel on the image in degrees Celsius (°C), $T_i$ is the surface temperature of the *i*th pixel in degrees Celsius (°C), *n* is the total number of all valid pixels in a suburban area, and $T_{coolj}$ is the surface temperature of the *j*th pixel in the cold surface area (vegetated land and water) in degrees Celsius (°C).

### 3.2.2. Retrieval of PM$_{2.5}$ Concentrations

At present, well-developed methods for retrieving the aerosol optical depth (AOD) from satellite remote sensing data include the Dark Target (DT) [58], Multiangle Implementation of Atmospheric Correction (MAIAC) [59,60], and Deep Blue (DB) [61] algorithms. The difference between these methods lies in how to remove the contribution of surface reflectance in the apparent reflectance of images, which also leads to their different applicability. The DT algorithm can be applied only to areas with a high vegetation density in spring and summer; otherwise, the accuracy is low [62]. Because the imaging time in this study is winter and the high-density urban area is dominated by buildings and roads and lacks dense vegetation, the DT algorithm is not suitable for this study. Furthermore, the image range required for the MAIAC method does not cover China and cannot be used in this study. The DB algorithm uses the strong atmospheric reflectance and weak surface reflectance characteristics in the dark blue band to remove surface reflectance contributions. Compared with the DT algorithm, the DB algorithm is suitable for more types of land surface, which is more widely used and is not affected by the seasons [63]. Therefore, this study selects the DB algorithm as the AOD retrieval method. The retrieval process is to first use the DB algorithm to retrieve the AOD from the Chinese Gaofen-1 satellite data and then establish a linear regression model between the observation data of PM$_{2.5}$ concentration and the AOD retrieval results to obtain the PM$_{2.5}$ concentration field. The specific retrieval process can be found in the literature [64,65].

### 3.2.3. Analysis of Wind Environment

According to relevant research [66], when implementing and building ventilation corridors, it is necessary to conduct a comprehensive wind environment assessment of the target city or region and have a macroscopic understanding of the location of compensation spaces, the potential available wind system and local circulation system. The urban wind system includes local and dominant winds [67]. The local wind refers to ventilation resources that are affected by factors such as the location distribution and local topographic characteristics, which are manifested as increasing wind speeds in urban areas under weak background wind conditions, such as valley winds and land–sea breezes. The dominant wind has airflow field characteristics that affect the region year-round, changes regularly with the seasons, and is affected by large-scale monsoon circulation. The dominant wind is a key index for evaluating ventilation corridors.

### Assessment of Dominant Wind Direction

Shanghai has a typical subtropical monsoon climate, and there are significant differences in the dominant wind direction in summer and winter. Therefore, it is necessary to further analyze the actual dominant wind direction characteristics in the study area. Wind rose diagrams in winter and summer are plotted, and the dominant wind directions are determined through statistical analysis of the meteorological observations from the state-monitored weather stations from 2000 to 2021.

Simulation of Local Wind Field

The local wind field in the study area is simulated using the WRF [50] model combined with the BEP [51] urban canopy model [68,69]. Due to the accelerating urbanization process and the failure to update the WRF land surface data in a timely and effective manner, there is a significant deviation between the actual observations of the meteorological environment and the simulation results of the WRF model [70–73]. Therefore, this study first extracts the urban impermeable surface information using the normalized difference impervious surface index (NDISI) developed by Xu [74] and refines the urban built-up areas into "low-density areas", "medium-density areas" and "high-density areas" according to their NDISI differences to optimize the land use data in the WRF model [75,76].

In this study, the wind field in Shanghai and its surrounding areas is simulated using the WRF (V3.9) model coupled with the BEP model. The simulation center is located at $120°33'$ E, $31°17'$ N. The simulation starts at 2:00 Beijing time on 20 February 2021, and ends at 8:00 on 22 February 2021. The WRF model uses a three-layer bidirectional nested grid partition scheme, which couples the BEP model in the third mode domain. The horizontal resolutions of the grid are 1 km, 3 km, and 9 km (from the inside out), and the grid is vertically divided into 50 layers, of which there are 22 vertical layers encrypted below 2 km, and the height of the first grid is approximately 10 m above the ground. A $0.25°$ global tropospheric analysis on a 6 h basis [77] provides the initial and boundary meteorological conditions to the mother domain. The refined GlobeLand30 land use data (Figure 4) are input into the model. The specific physical scheme settings of the WRF simulation can be found in the research of Xu et al. [78] and Xu et al. [79].

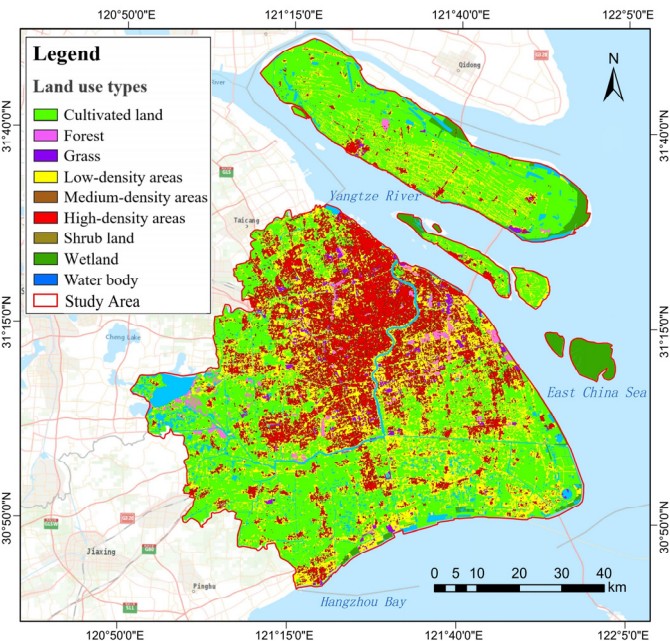

**Figure 4.** Distribution of land use types in Shanghai.

By calculating the Bias (observed value minus simulated value), root mean square error (RMSE), and correlation coefficient (Corr) and its significance test between the simulation results and the observations, the temperature at 2 m ($T_2$), the wind speed at 10 m ($WS_{10}$), and the relative humidity (RH) output by the WRF model are used for simulation performance evaluation. In the verification statistics of meteorological elements, each station uses the three-hour average value calculated from the 48 h of data output by the model and 48 h of observation data for verification. Verification results at Baoshan station (BSs, 58362), Xujiahui station (XJHs, 58367), are shown in Table 3. In the tables, ** indicates that the *p*-value is less than 0.01, which means it is very significant. The statistics reveal that the

model simulations agree well with the observations, so the simulation results used in this study can be considered reliable.

**Table 3.** Validation statistics of WRF output results.

| Station | $T_2$ (°C) | | | RH (%) | | | $WS_{10}$ (m/s) | | |
|---|---|---|---|---|---|---|---|---|---|
| | Bias | RMSE | Corr | Bias | RMSE | Corr | Bias | RMSE | Corr |
| BSs | −2.51 | 2.14 | 0.85 ** | 11.32 | 14.24 | 0.72 ** | 0.11 | 0.73 | 0.76 ** |
| XJHs | −2.12 | 1.78 | 0.88 ** | 12.65 | 15.33 | 0.79 ** | 0.27 | 0.88 | 0.73 ** |
| Average | −2.32 | 1.96 | 0.87 | 11.99 | 14.79 | 0.76 | 0.38 | 0.81 | 0.75 |

** Indicates statistically very significant correlation ($p < 0.01$).

## 4. Results

### 4.1. Characteristics of the Wind Environment in Shanghai

4.1.1. Wind Rose Diagrams in Winter and Summer

According to the wind rose maps (Figure 5), the primary wind direction during the summer in Shanghai is east–northeast, accounting for 15% of the total frequency. Additionally, the frequencies of south–east, east–southeast, and east winds are relatively high at 11.5%, 10.5%, and 10%, respectively, contributing to a total frequency of 55% from the northeast to the southeast. Thus, Shanghai experiences predominantly easterly winds in the summer. On the other hand, during the winter in Shanghai, there is little difference in the frequencies of east–northeast, northeasterly, northerly, west–northwest, and westerly winds, which account for 13%, 12%, 10%, 9.5%, and 9% of the total frequencies, respectively. The total wind frequency from the northwest to the northeast is 39%, indicating a dominance of northerly winds in winter. To summarize, Shanghai exhibits a significant contrast in the dominant wind direction between winter and summer seasons. Therefore, two wind directions will be considered when implementing and building UVCs: summer prevailing wind (easterly winds); and winter prevailing wind (northerly winds).

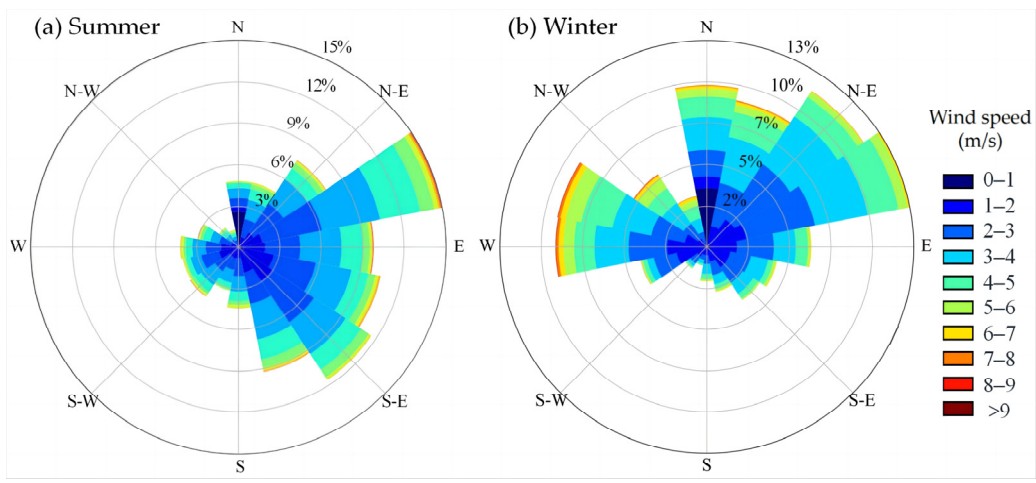

**Figure 5.** Wind rose maps of Shanghai in summer (**a**) and winter (**b**).

4.1.2. Simulation Results of the Wind Field

A 10-m-high wind field map of the innermost layer of the nested layer is drawn by running the NCL postprocessor of WRF, and six time node maps are drawn daily to analyze the changes in the daily wind field in the study area and the local circulation distribution characteristics of the small area. Wind field maps (Figure 6) are generated for six time nodes (2:00, 6:00, 10:00, 14:00, 18:00, and 22:00 Beijing time) on 21 February 2021.

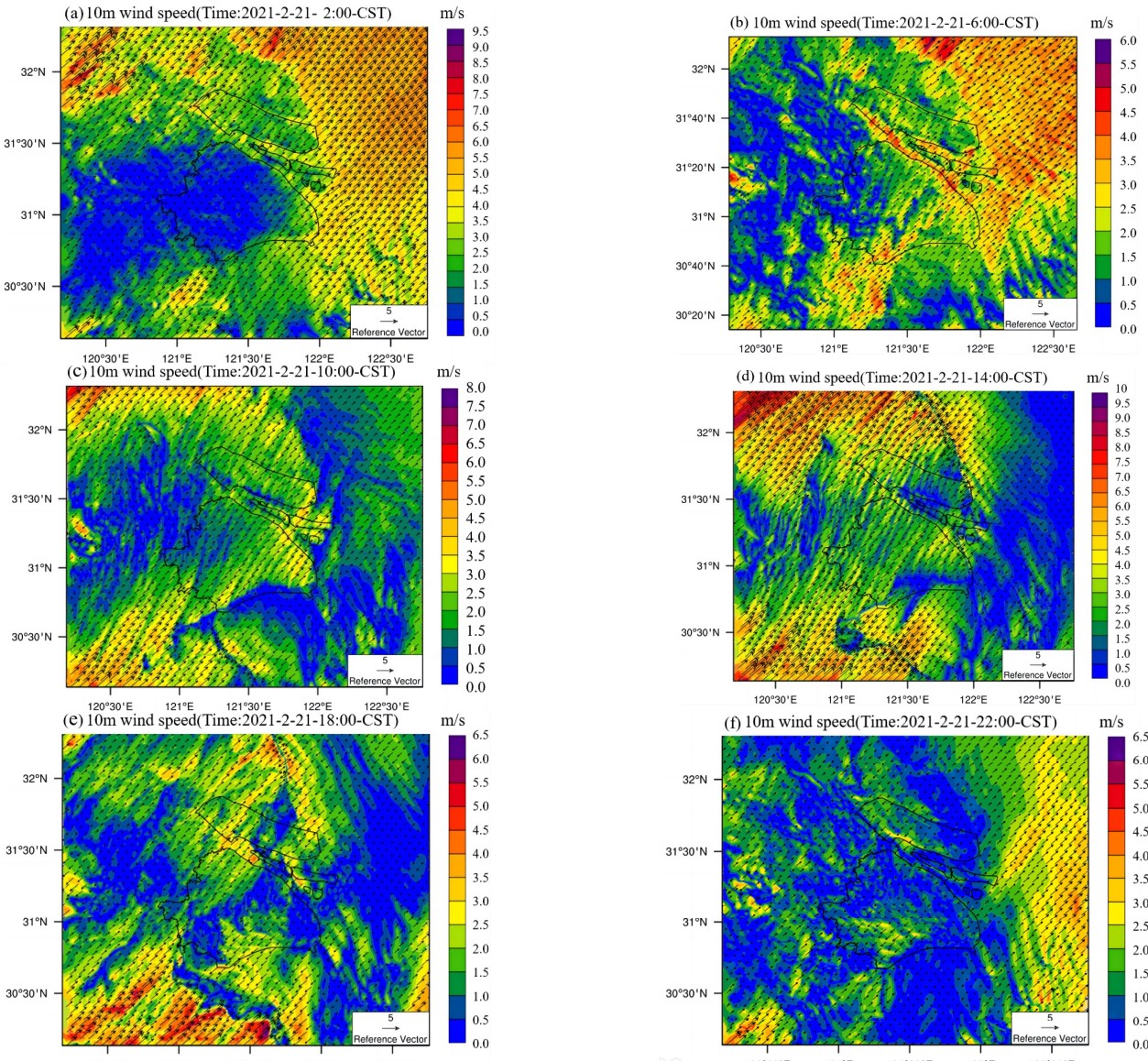

**Figure 6.** The simulation results of wind field in Shanghai and its surrounding areas.

By observing the changes in the wind environment at the various time nodes during the simulation period, the regional background wind is shown to be a southwest wind only, and the overall airflow moves from the southwest (inland) to the northeast waters, which is consistent with the measurements of the Shanghai Meteorological Station data on that day; that is, the dominant wind direction on 21 February 2021 is a southwest wind.

Under the effect of background wind, there are also local airflow exchanges and local circulation, especially land and sea breeze. It can be seen from Figure 6 that from 2:00 to 14:00, under the joint action of the land wind and background wind moving in the same direction, the dominant southwest wind in Shanghai gradually increases from an initial average wind speed of 1 m/s to 3.5 m/s. Then, at approximately 14:00, there is a sea breeze coming from the opposite direction to the background wind direction on the east coast of Shanghai. Subsequently, the sea breeze from the northeast gradually replaces the background wind and becomes the dominant wind direction of Shanghai, which lasts until approximately 22:00. In addition, as shown in Figure 6, high-density buildings in the central urban area act as barriers, resulting in lower turbulence and wind speeds compared to other areas. In these low wind conditions, the urban interior is highly susceptible to a strong urban heat island effect.

### 4.2. UHI Effect in Shanghai

#### 4.2.1. Characteristics of LST Distribution

To analyze the UHI effect in Shanghai, Landsat 8 images captured during sunny summer days are selected for this study, and LSTs are derived using the aforementioned methods. Based on this, the relationship between land use types and land surface temperature is analyzed to reveal the spatial distribution characteristics of land surface temperature in Shanghai. By overlaying the LST map with the land use type map (Figure 4), a zoning statistical method is used to generate the spatial distribution map of average LSTs for different land use types (Figure 7). The average LST of different land use types increases in the following order: sea, water bodies, wetlands, bushlands, cultivated lands, forests, grasslands, low-density urban areas, medium-density urban areas, and high-density urban areas. The average LST in the high-density central urban area is the highest at 41.1 °C, followed by the medium-density urban area at 39.6 °C, and the low-density urban area at 36.6 °C. On the other hand, the average LST of undeveloped urban areas, including water bodies, wetlands, bushes, cultivated land, forests, and grasslands, is significantly lower compared to developed urban areas. This indicates that suburban woodlands, as well as large bodies of water and green spaces within the city, are crucial sources of cool air for mitigating the urban heat island effect. They can serve as important ventilation inlets for UVCs.

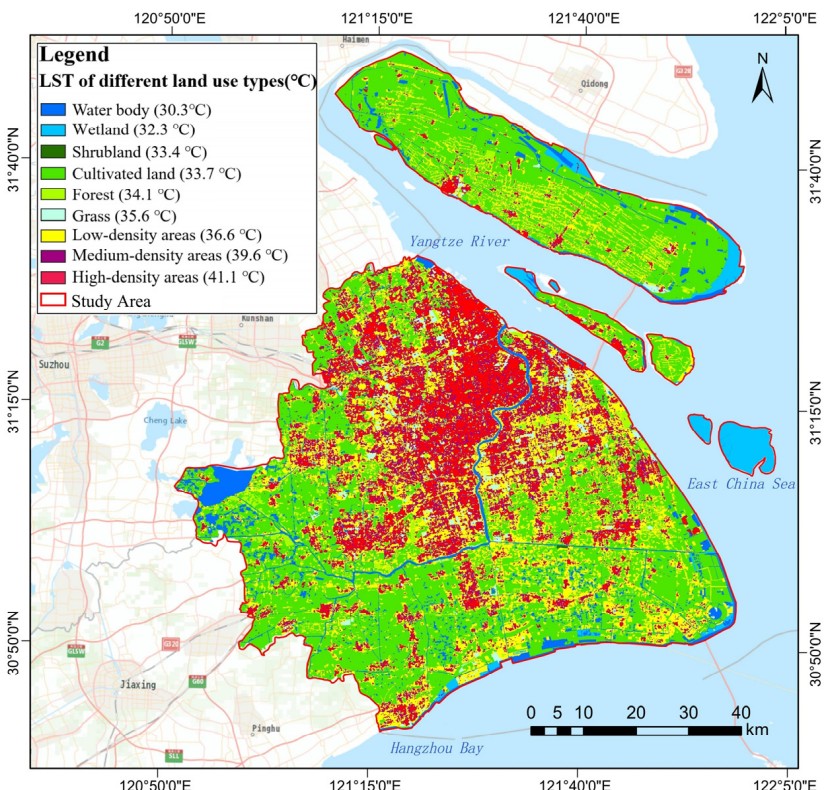

**Figure 7.** Spatial distribution of LST of different land use types.

#### 4.2.2. Characteristics of UHII Distribution

As shown in Figure 8, the distribution characteristics of the UHII in the whole study area are relatively obvious. The UHII gradually shows a radiant pattern, from high level in the central urban area (the strong heat island core area) to low level in the periphery. Moreover, the urban areas in the east–west direction of the Huangpu River show an asymmetrical heat islands distribution pattern; that is, the strong heat island areas in Shanghai are mainly concentrated west of the Huangpu River. To the east of the Huangpu River, only the east bank of the Huangpu River and some commercial and trade zones, industrial parks, and high-density residential areas in Pudong are strong heat island areas.

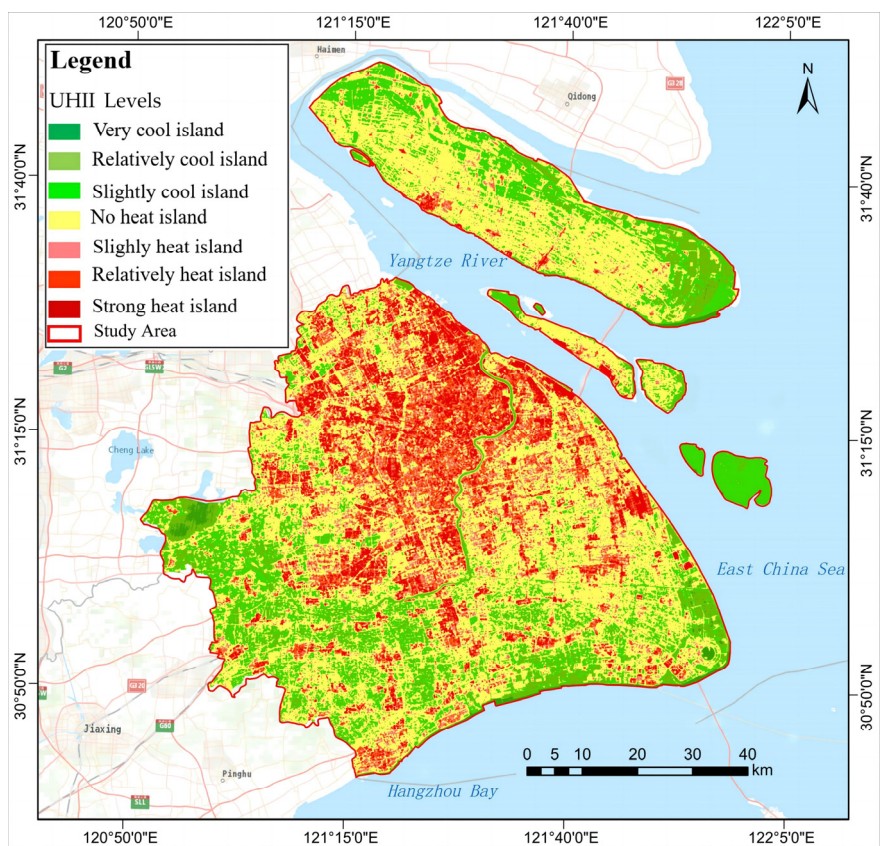

**Figure 8.** Spatial distribution of UHII in Shanghai.

4.2.3. Identification of Action and Compensation Spaces according to Their Thermal Field

According to the spatial distribution of the cool and heat island areas, the action and compensation spaces are identified. As compensation spaces, low-LST areas can be roughly divided into cold air-generating areas, mainly including suburban woodlands, cultivated lands, large parks, large areas of water, sea areas around the city, and other large green spaces. The heat island areas, mainly comprised of the industrial and commercial areas and the dense residential complexes in the town, serve as the action spaces for the ventilation corridors.

As shown in Figure 9, A1–A27 are the compensation spaces in Shanghai, including Dianshan Lake (A1), Jiabei Country Park (A2), Gucun Park (A3), farmland in the south of Taxin East Road (A4), cultivated land near Hutong Railway in Jiading District (A5), cultivated land and waters near Cao'an Road (A6), intersection of Shanghai-Changzhou Expressway and Shanghai Bypass Expressway (A7), Changfeng Park (A8), Gonghe Park (A9), Sun Island Tourist Resort (A10), cultivated land and waters near Jiyou Road (A11), Daning Tulip Park (A12), Sheshan National Forest Park (A13), woodland near South Third Highway (A14), Minhang Sports Park (A15), People's Park (A16), cultivated land and waters near Xingtuan Road and Yingli Road (A17), Pujiang Country Park (A18), woodland north of Dongjia Road (A19), woodland south of Jinxiu East Road (A20), cultivated land around Guanguang Road (A21), cultivated land near Liangdian North Road (A22), woodland around Huadong Road (A23), Shanghai Gulf National Forest Park (A24), Wildlife Park (A25), cultivated land near Xinni Road (A26), and green space along Huiping South Road (A27).

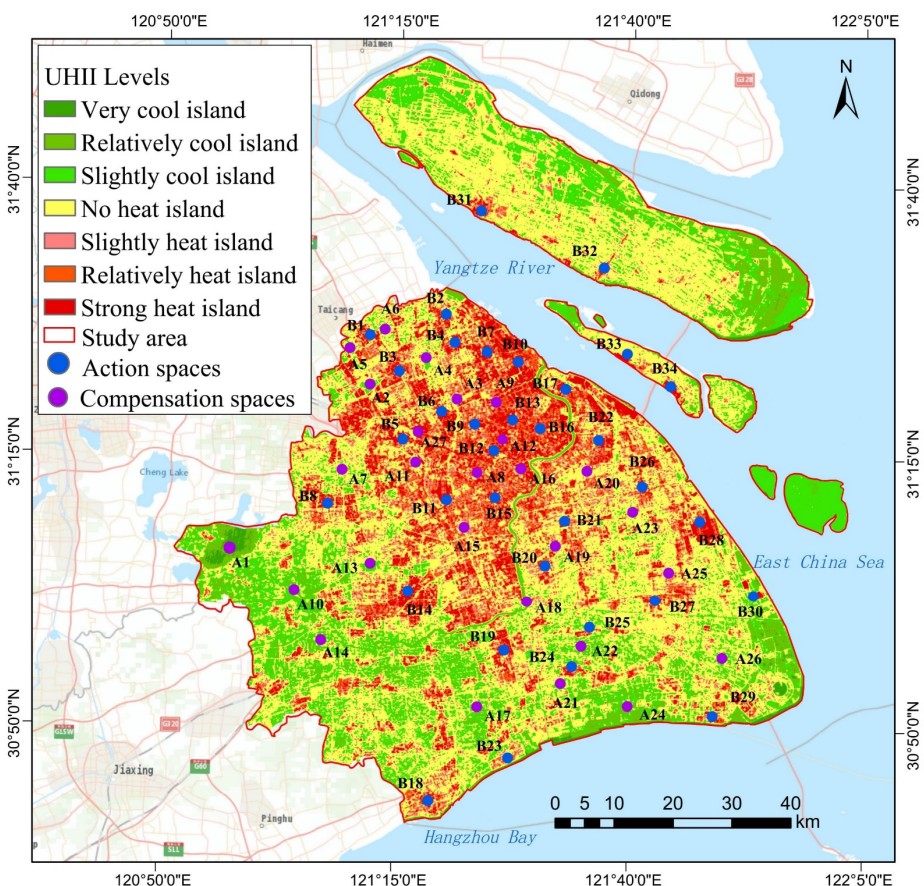

**Figure 9.** Compensation and action spaces of ventilation corridor determined according to thermal field.

B1–B34 are the action spaces, including the industrial area near Beihe Road (B1), the industrial area around Panchuan Road (B2), the intensive complex near Tacheng Road (B3), the commercial area near Luoxin Road (B4), the industrial area around Xinghua Road (B5), the industrial area on Fengxiang Road (B6), the industrial area between Yuanhe Road and Chunhe Road (B7), the industrial area in the western section of Songze Avenue (B8), the intensive complex along Nanda Road (B9), the industrial area around Tieli Road (B10), Hongqiao International Airport (B11), the intensive complex near Jiaotong Road (B12), Beijiao Railway Station (B13), the industrial area near Rongle Road (B14), the commercial area and university campus near Yishan Road (B15), Pentagon Business District (B16), the industrial area near Gangcheng Road (B17), commercial area and university campus near Weiqing Road (B18), the industrial area in the eastern section of Shanghai–Hangzhou Highway (B19), the industrial and commercial areas around Hengnan Road (B20), the industrial area around Kangwu Road (B21), the industrial area near Shenjiang Road (B22), the industrial area in the south of Nanyinhe Road (B23), the industrial area near Pinxing Road (B24), the industrial area near Litai Road (B25), the industrial and commercial areas near Lihang Road (B26), the intensive complex along Huicheng Road (B27), Pudong International Airport (B28), the industrial area around Canghai Road (B29), the renewable energy utilization center of Laogang (B30), the intensive complex near Ximen Road, Chongming District (B31), the industrial area around Baozhen South Road (B32), the industrial area of Hongxing Village, Chongming District (B33), the shipyard near Jiangnan Avenue, Changxing Island (B34).

*4.3. PM$_{2.5}$ Pollution in Shanghai*

4.3.1. Characteristics of the PM$_{2.5}$ Concentration Field

According to the DB algorithm, the AOD during the winter in Shanghai is inverted, as shown in Figure 10. Since AOD is a dimensionless physical quantity used to measure the relative concentration of aerosols, it cannot directly reflect the air pollution situation in the study area. In order to reveal the relationship between air pollution and AOD, this study compares the inverted AOD values with the measured PM$_{2.5}$ concentrations at monitoring stations in Shanghai. Correlation analysis and linear regression are performed on the measured PM$_{2.5}$ concentrations and corresponding AOD inversion values at 18 monitoring stations in Shanghai at 10:52 on 21 February 2021 (Beijing time). The regression model is shown in Figure 11. The results of the correlation analysis show that the Pearson correlation coefficient between AOD and the measured PM$_{2.5}$ concentration is 0.85, with a significant correlation level of 0.01 (two-tailed). This indicates a strong positive correlation between the measured PM$_{2.5}$ concentration and the AOD. Using this model, the PM$_{2.5}$ concentration field in Shanghai is further inverted, as shown in Figure 12.

To verify the retrieval accuracy of the PM$_{2.5}$ concentration, the simulation and measured values of the PM$_{2.5}$ concentration are statistically analyzed. A statistically significant correlation is observed ($R^2 = 0.72$, $p < 0.01$), the root mean square error (RMSE) is 1.58 μg/m$^3$, and the mean absolute error (MAE) is 0.37 μg/m$^3$. Therefore, this model can better simulate the distribution of PM$_{2.5}$ concentration.

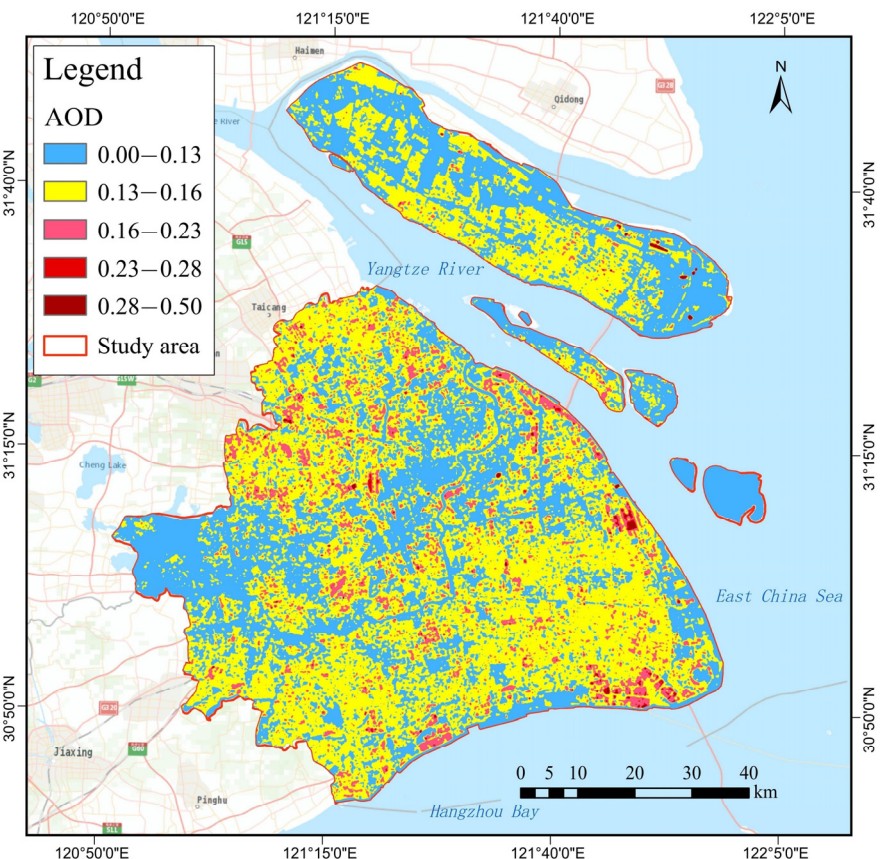

**Figure 10.** Retrieval results of AOD in Shanghai.

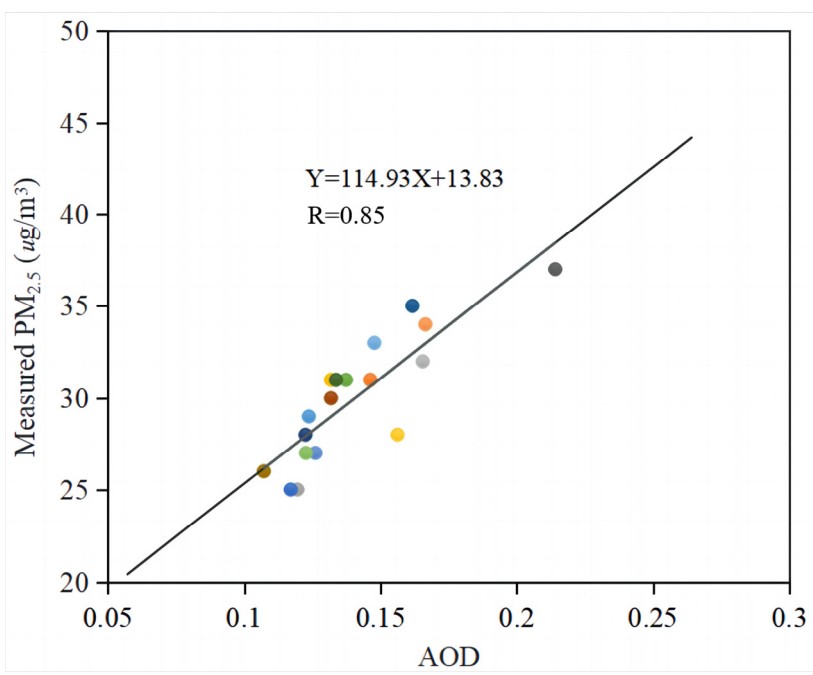

**Figure 11.** Linear regression between AOD and measured PM$_{2.5}$ concentration ($p < 0.01$).

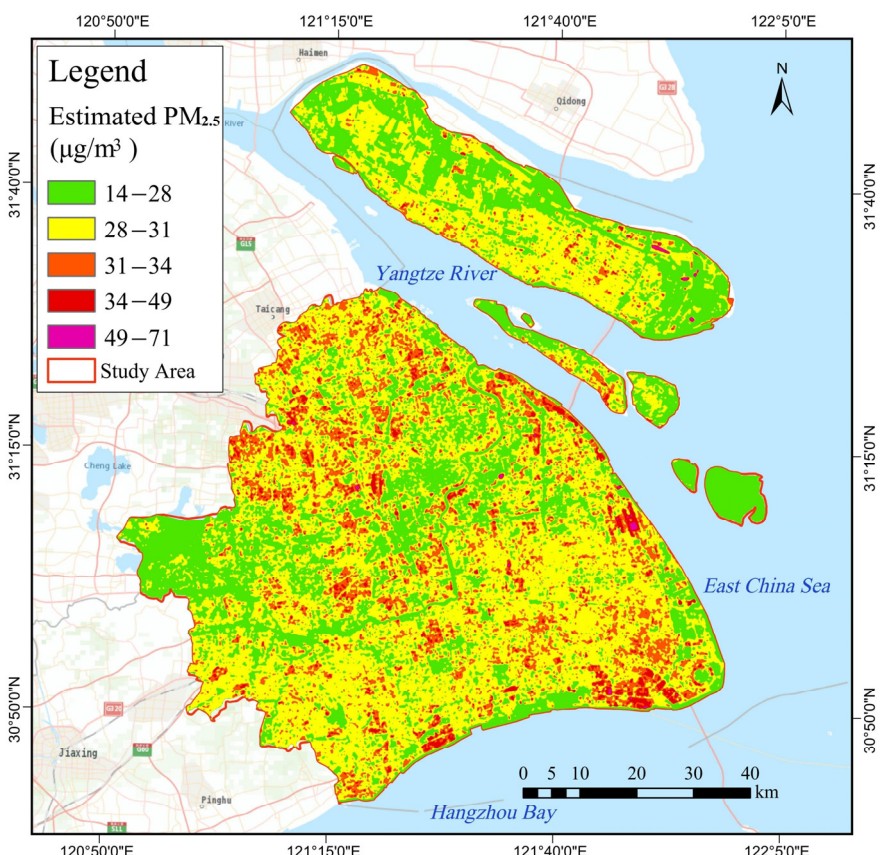

**Figure 12.** Distribution of PM$_{2.5}$ concentration in Shanghai.

Figure 12 shows that the PM$_{2.5}$ concentration field in Shanghai has an obvious inter-phase distribution pattern of high and low PM$_{2.5}$ concentration fields. In contrast to the distribution of thermal field (Figure 8), the PM$_{2.5}$ concentration is relatively low in the central urban areas east and west of the Huangpu River, including the Yangpu, Hongkou,

Jing'an, Changning, Xuhui and Huangpu districts, with an average of less than 26 $\mu g/m^3$. On the other hand, the $PM_{2.5}$ concentration in the Baoshan District, Jiading District, and the northeastern Qingpu District adjacent to Jiangsu Province is relatively high, and the average concentration exceeds the Shanghai average level of 31 $\mu g/m^3$. There are also many high-$PM_{2.5}$-concentration fields along the southeast coast, of which the southern coastal area of the Jinshan and Fengxian districts and the southeastern coastal area of the Pudong New Area are the most prominent, with an average concentration of $PM_{2.5}$ exceeding 35 $\mu g/m^3$.

By comparing and analyzing Figures 9 and 12, it is evident that certain cold air compensation spaces have high $PM_{2.5}$ concentration, such as Jiabei Country Park and cultivated land in the northern Jiading District. A previous study by Han et al. [39] has revealed that if the urban compensation spaces are located in areas with high concentrations of pollutants, the polluted cold air will be introduced into the central urban area along ventilation corridors, driven by prevailing wind. Therefore, it is essential to evaluate the air quality of compensation spaces during the planning of UVCs to prevent the exacerbation of air pollution in the central urban area and the deterioration of the urban microclimate.

4.3.2. Identification of Action and Compensation Spaces according to $PM_{2.5}$ Concentration Field

Based on the theory of local circulation patterns proposed by Kress [41], the action space is the area with the most concentrated and serious UHI effect and air pollution. The compensation space is where the UHI effect is relatively weak, and the air pollution is relatively light. Previous studies on UVCs have focused on analyzing the spatial distribution of thermal field to determine the action and compensation spaces of UVCs. However, they failed to analyze the spatial distribution of air pollution. To fill this gap, this study initially identifies locations of compensation spaces according to thermal field and assesses their air quality according to $PM_{2.5}$ concentration field.

The compensation space and the action space are determined by the superposition analysis of Figures 9 and 12, and the results are shown in Figure 13. C1–C22 are the compensation spaces with low $PM_{2.5}$ concentrations, including Jiabei Country Park (C1), Intersection of Shanghai–Changzhou Expressway and Shanghai Bypass Expressway (C2), Fanglin Road (C3), Gucun Park (C4), Dianshan Lake (C5), Jiyou Road (C6), Gonghe Park (C7), Sun Island Tourist Resort (C8), Sheshan National Forest Park (C9), Changfeng Park (C10), Daning Tulip Park (C11), the area near South Third Highway (C12), Minhang Sports Park (C13), People's Park (C14), the area between Yingli Road and Xingtuan Road (C15), Pujiang Country Park (C16), Jinxiu Road (C17), the area near Yanqian Road (C18), south section of Huadong Road (C19), Shanghai Gulf National Forest Park (C20), Wildlife Park (C21), and farmland in the south of Taxin East Road (C22).

The action spaces are the areas with high $PM_{2.5}$ concentrations, as shown in Figure 13 D1–D24, including the area near Beihe Road (D1), Panchuan Road (D2), the intensive complex near Luoxin Road (D3), the area near Cao'an Road (D4), the industrial area between Chenxiang Road and Fengxiang Road (D5), the area between Panjing Road and Jiangyang North Road (D6), Songze Avenue (D7), the area between the Outer Ring Expressway and Qilianshan Road (D8), the intensive complex near Tieli Road (D9), Hongqiao International Airport (D10), the area between Fangsi Highway and Shenhai Expressway (D11), the area between Rongle East Road and Hukun Expressway (D12), Gangcheng Road (D13), the intensive complex between Kunyang Road and Shenjiahu Expressway (D14), the industrial area south of Weiqing Road (D15), the area near the eastern section of Shanghai-Hangzhou Highway (D16), Kangwu Road (D17), Shenjiang Road (D18), the industrial area in the south of Nanyinhe Road (D19), the area around Daye Highway (D20), Pudong International Airport (D21), middle section of Lianggang Avenue (D22), the southern area of Lingang Avenue (D23), and the industrial area between Jiangnan Avenue and Panyuan Highway, Changxing Island (D24).

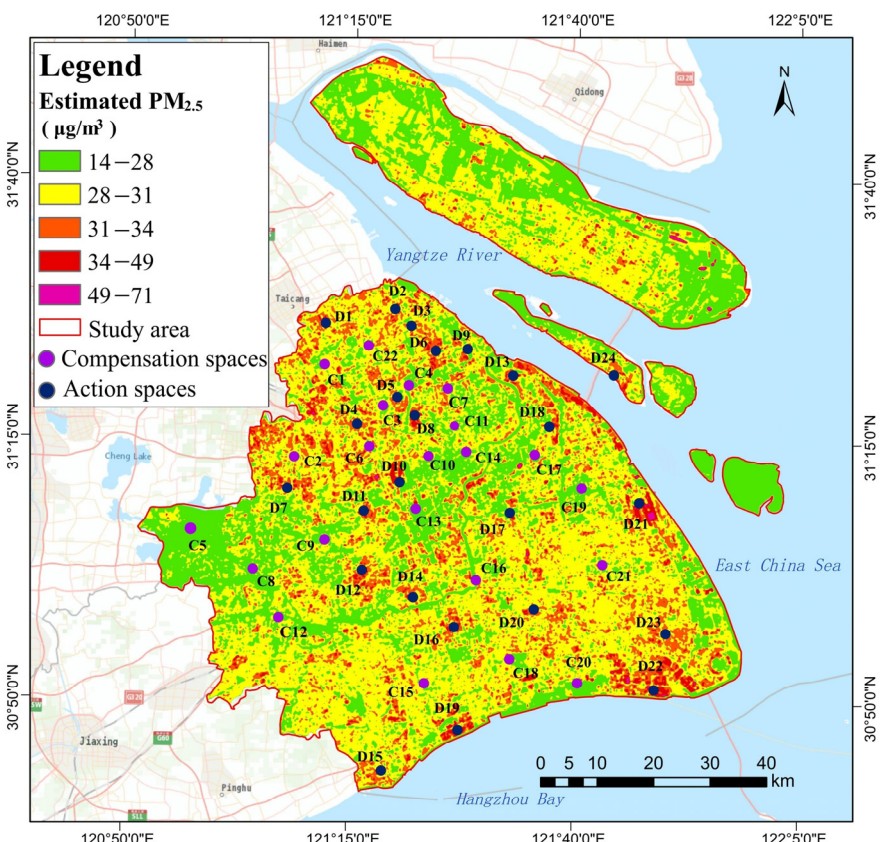

**Figure 13.** Compensation and action spaces of ventilation corridor determined according to PM$_{2.5}$ concentration field.

### 4.4. Optimization and Planning of UVCs

#### 4.4.1. Planning and Design of UVCs

In contrast to the previous design methods of urban-scale ventilation corridors, the spatial distribution of the action space and compensation space is comprehensively analyzed based on the urban thermal field and PM$_{2.5}$ concentration field. The compensation space is identified as the area where the heat island effect is relatively weak and the air pollution is relatively light, which is generally the starting point in a ventilation corridor design. On the other hand, the action space is identified as the area with the most concentrated and serious heat island effect and air pollution and is generally the end point in a ventilation corridor design. The direction of ventilation corridors is determined on the basis of the dominant wind direction in winter and summer and the simulation result of the wind field in Shanghai. By connecting the compensation and action spaces, seven potential first-class ventilation corridors (represented by I$_1$, I$_2$, I$_3$, I$_4$, I$_5$, I$_6$, and I$_7$) and nine potential secondary ventilation corridors (represented by II$_1$, II$_2$, II$_3$, II$_4$, II$_5$, II$_6$, II$_7$, II$_8$, and II$_9$) are constructed, as shown in Figure 14.

Among them, the first-class ventilation corridors are mainly designed based on the dominant wind direction in summer and winter and the wind field simulation. They run through the study area from southwest to northeast, thus forming the "urban ventilation artery" in Shanghai. Furthermore, based on the prevailing southeasterly wind direction in summer and the northwesterly wind direction in winter, nine secondary ventilation corridors are built using primary and secondary traffic trunk roads, rivers and other continuous open areas in the urban area to connect the parks, lakes, green spaces, and other cold sources in the urban area. These secondary ventilation corridors can transport fresh air from the first-class air channels into the action spaces to further improve the urban ventilation conditions.

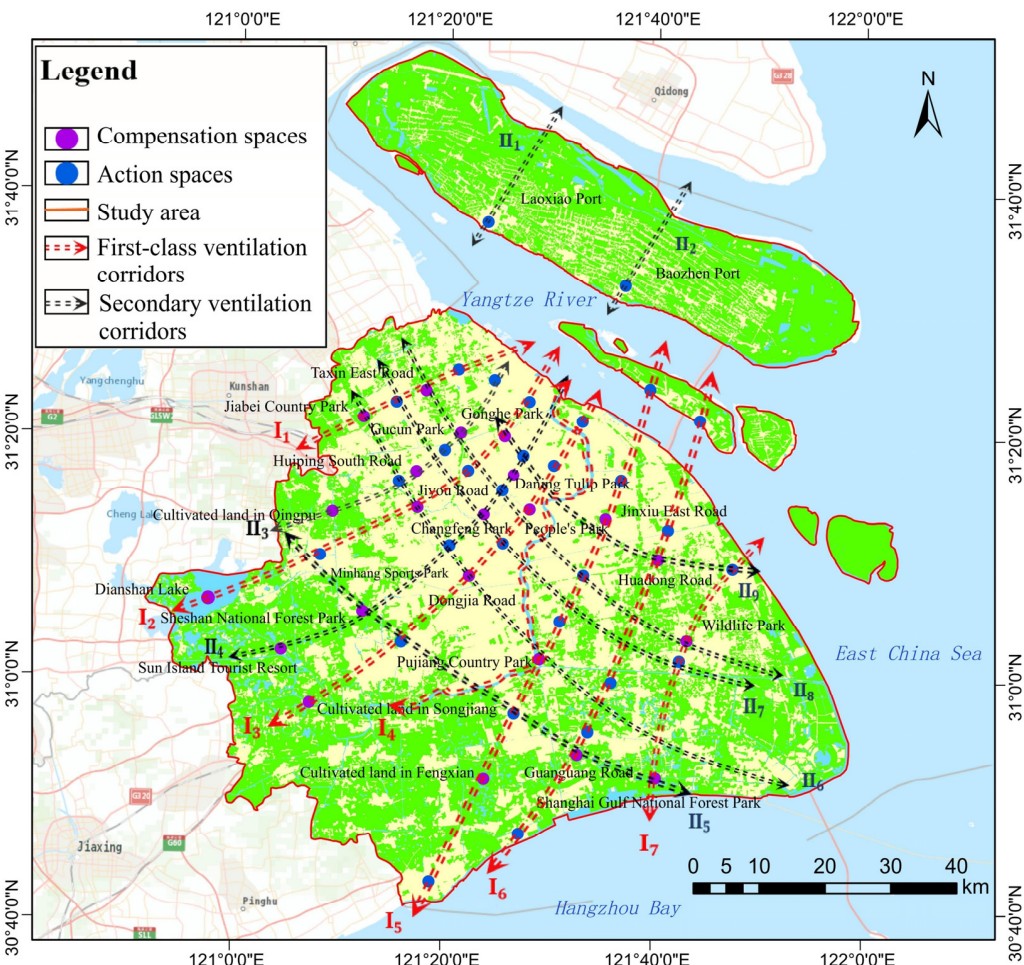

**Figure 14.** Construction of ventilation corridors in Shanghai.

## 4.4.2. Optimization and Planning Strategies for Ventilation Corridors

Shanghai's ventilation corridors consist of first-class ventilation corridors and secondary ventilation corridors, forming a network to promote air circulation in urban areas, which can effectively channel the fresh air around the city to the central urban area. Based on the above design of ventilation corridors, to better utilize the role of ventilation corridors, it is also necessary to propose planning strategies and suggestions to maintain and improve the urban ventilation potential [80], and the optimization strategy is summarized in Table 4.

## 4.5. Comparison of Different Methods

Figure 15 compares the Shanghai central areas (Changning, Xuhui, and Jing'an) results using various methods. The modified RS analysis, the original RS analysis, and the LCP analysis consist of the extraction results under the SW background wind direction. The UVCs identified using various methods are sufficiently consistent. However, Figure 15c shows that the GIS spatial analysis does not sufficiently reflect VC patterns under different wind directions. Further, the results based on the original RS analysis, LCP analysis and GIS spatial analysis are similar. They cannot all represent the spatial distribution of air pollution in VCs. In Figure 15a, the compensatory spaces, such as Minhang Cultural Park and Hongqiao Golf Course, are located in areas with high PM$_{2.5}$ concentrations. If a compensation space is located in a region with high pollutant concentrations and is influenced by prevailing winds, polluted cold air can be drawn into the central urban area through VCs, leading to worsening air pollution in residential areas.

**Table 4.** Control and management strategies for ventilation corridor planning.

|  | **First-Class Ventilation Corridor** | **Secondary Ventilation Corridor** |
|---|---|---|
| Air inlet | Large-scale waters, sea areas, large parks, open green spaces | Rivers, parks, green space and roads |
| Length of planned ventilation corridor | ≥30 km | ≥15 km |
| Width of planned ventilation corridor | ≥200 m | ≥50 m |
| Width of obstacles perpendicular to air flow | ≤10% of the corridor's total width | ≤20% of the corridor's total width |
| The angle between planned corridors and the prevailing wind direction | ≤30° | ≤45° |
| Proportion of construction land within the corridor | ≤20% | ≤25% |
| Remarks on management and control | The planning and construction of ventilation corridors should be integrated with the construction of urban green corridors, large parks, and other green spaces, as well as rivers, urban main roads, etc. It is not advisable to demolish and rebuild on a large scale. Strictly control the construction scale, prohibit high-density construction and development, maintain the openness of space, and minimize the building coverage as much as possible. Establish a long-term mechanism for controlling the height, density, and layout of buildings on ventilation corridors. More greening in built-up areas to further improve the ventilation power of the corridors. | |

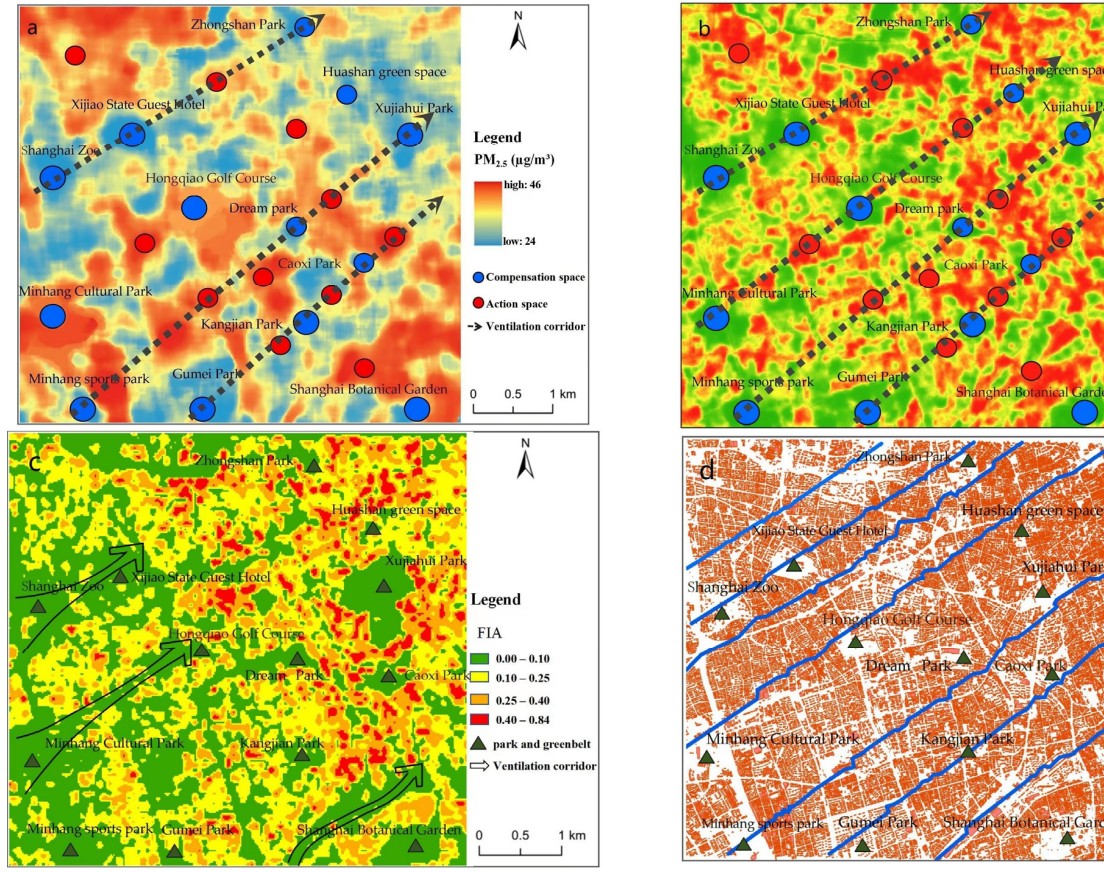

**Figure 15.** UVCs extracted using different methods: (**a**) modified RS analysis; (**b**) original RS analysis; (**c**) GIS spatial analysis; and (**d**) LCP analysis.

## 5. Discussion

### 5.1. Advantages

This study proposes a new method for urban-scale ventilation corridor optimization and planning that combines RS techniques and wind environment numerical simulations. The results of the modified RS analysis method showed improvement compared to those of the current RS analysis method. The proposed method has several advantages over the traditional method of extracting UVCs. First, compared with the wind tunnel and CFD methods, the proposed method can process larger amounts of data and is more suitable for urban-scale VC analysis. Second, compared with the GIS spatial analysis method, the proposed method can better represent the distribution characteristics of VCs under different wind directions. Third, compared with the LCP analysis method, the results of the proposed method can cover the entire study area and are not limited by building information data. Finally, compared with the traditional method, the proposed method adds to the process of air quality assessment. When determining the compensation space, an assessment of the air cleanliness in the compensation space is conducted, which can ensure the cleanliness of the air in the compensatory space to avoid aggravation of the air pollution in the central urban area.

### 5.2. Protection Strategies for Ventilation Corridors

In the study area, ventilation corridors exhibit different spatial patterns. Exploring these patterns and proposing corresponding optimization measures is crucial for sustainable urban development [66]. As shown in Figure 14, The first-class ventilation corridors consist of ecological functional areas, large-scale park green spaces and low-intensity development zones. They are not only important air guide channels under dominant wind conditions but also air exchange spaces between the land–sea breezes and the city. Their climate and environmental values are extremely high.

First, the ecological protection of compensatory spaces should be strengthened. For example, the red line of ecological protection in Jiabei County Park, Dianshan Lake, Shanghai Gulf National Forest Park, and Shanghai Wildlife Park should be strictly controlled. These ecological green spaces should be strictly protected from encroachment by urban development to protect the area of the air conditioning reservoir.

The spatial pattern around the green corridor should be optimized. For example, the layout of buildings around Pujiang County Park in Minhang District should be strictly controlled to create a three-dimensional layout of buildings with open spaces, a moderate density, large height differences, and rich interface changes. The height of the building should be limited to 15 m and the density of the building should not exceed 20% to avoid blocking corridor ventilation.

The environmental protection of water bodies near the ventilation corridor should be strengthened. For example, the water quality of the Huangpu River should be better protected, clean air sources should be guaranteed, and a green landscape belt should be constructed along the river. Parallel "points" of green spaces should be installed on both sides of the Huangpu River at 1 km intervals, and ribbon-like green spaces should be added perpendicular to the river and main traffic roads to form a gridded green belt system.

The spatial layout on both sides of the road ventilation corridor should be reasonable. For example, for road-type air ducts such as Guanguang Road in Fengxian District and Huadong Road in Pudong New District, it is necessary to control the height, density, arrangement, and orientation of buildings around the roads [81]. The height-to-width ratio of buildings on both sides of the air ducts should not be greater than 0.5, and the degree of openness should not be less than 40%. The arrangement of buildings should be a combination of staggered and inclined arrangement to create a relatively continuous open space and ensure the ventilation effect of the air duct.

### 5.3. Application in Urban Planning

This study demonstrates the disparities in spatial distribution characteristics between the urban heat field and $PM_{2.5}$ concentration field. If the urban compensation space is situated in areas with high pollutant concentrations, it will introduce polluted cold air into the central urban area through the urban ventilation corridor, exacerbating air pollution levels. Although many cities around the world have carried out research on ventilation corridors, and more cities will implement this practice in the future, the lack of assessment regarding cold air cleanliness remains a common issue in urban ventilation corridor planning. This oversight can result in less effective ventilation corridors, or even worsening air pollution in the central city. Therefore, in order to achieve scientifically sound ventilation corridor planning, it is crucial to first identify the compensation space and action space and assess the air cleanliness in the compensation space.

The proposed method can effectively identify urban-scale ventilation corridors, which has great potential for application in urban planning. First, this method can visually characterize the spatial distribution characteristics of urban air pollution and urban heat islands. It can be used to identify compensation spaces and acting spaces. By identifying these areas, urban environmental improvement funds can be maximally utilized. Second, the proposed method can provide decision support for UVC planning. Planners can select ventilation-friendly areas as data support for UVC planning. These selected areas typically possess weak UHI effects and high air cleanliness. Thirdly, the proposed method provides a comprehensive analysis approach for urban form development. It not only analyzes ventilation condition but also considers the UHI effect and air pollution condition. This method improves current UVC planning schemes, which can improve the urban microclimate environment. Finally, the proposed method can be used to evaluate the effect of UVCs on improving the microclimate environment in various cities, aiding in the exploration of sustainable urban development strategies.

### 5.4. Limitations and Future Work

This study presents a comprehensive method for optimizing and planning ventilation corridors at an urban scale, which combines LST retrieval, $PM_{2.5}$ concentration retrieval, and WRF simulation. However, more research is needed in the future. Specifically, while this study analyzed $PM_{2.5}$ concentration field to evaluate air cleanliness in the compensatory space of the ventilation corridor, the spatial distribution of other air pollutants such as $PM_{10}$, $O_3$, and $SO_2$ remains unclear. Therefore, future research should include analyses of these pollutants. Moreover, this study categorized urban built-up areas into "low-density", "medium-density", and "high-density" areas based on their NDISI differences to optimize land use data in the WRF model. To further improve the accuracy of simulations, more detailed and precise land use and building information data are needed in order to update the mesoscale meteorological models and increase analog resolution to street-level.

### 6. Conclusions

UVC has been demonstrated to play a crucial role in mitigating the UHI effect and reducing air pollution. This study presents a novel method for analyzing UVCs by integrating several factors, including LST retrieval, $PM_{2.5}$ concentration retrieval, wind field simulation, and dominant wind characteristics within urban areas. Using Shanghai as a representative example of a modern urban environment, seven potential primary ventilation corridors and nine potential secondary ventilation corridors are established. Additionally, more reasonable and detailed ventilation schemes are proposed, taking into account the distribution patterns of the ventilation corridors. This information offers a scientifically based evaluation of urban ventilation, which can support local governments in their urban planning endeavors.

Compared with wind tunnel testing, CFD simulation, GIS spatial analysis, and LCP-based UVC identification methods, the modified RS analysis technique possess faster data processing speeds, multi-directional dynamic simulation capabilities, and broader results

coverage. What is more, this proposed method adds to the air quality assessment process and increases the scientific accuracy and reliability of the final results compared with the original UVC analysis methods. A case study in Shanghai verified the practical application of this method. The study results demonstrate that this proposed method can help planners connect urban ventilation knowledge with urban planning needs for improving air quality, mitigating UHI effects, and promoting sustainable development of the urban microclimates.

However, further research is needed in the future. For example, more detailed and accurate land use and building information data are needed in order to update mesoscale meteorological models to improve the accuracy of WRF simulations. Moreover, future research endeavors should focus on improving the techniques used to evaluate the effects of UVCs, for example, by conducting comparative observational experiments that analyze the ventilation effects of UVCs in typical weather conditions and conducting extensive statistical analysis of VCs in different cities to explore patterns of urban sustainable development.

**Author Contributions:** D.G. proposed the methodology, data calculation, result analysis, and discussion. X.D. and L.Z. contributed to the data collection, and analysis. All authors have read and agreed to the published version of the manuscript.

**Funding:** The work described in this paper is funded by the National Natural Science Foundation of China (Grant No. 41501508).

**Institutional Review Board Statement:** Not applicable. Our study did not involve humans or animals.

**Informed Consent Statement:** Not applicable. Our study did not involve humans or animals.

**Data Availability Statement:** The $PM_{2.5}$ concentration data come from the National Urban Air Quality Real-Time Release Platform of China National Environmental Monitoring Center (https://air.cnemc.cn:18007/ (accessed on 18 March 2022)). Meteorological observation data come from China Meteorological Administration (http://data.cma.cn/ (accessed on 18 March 2022)). The Landsat 8 sensor data are obtained from the official website of the United States Geological Survey (USGS) (https://earthexplorer.usgs.gov/ (accessed on 18 March 2022)). The Chinese Gaofen-1 satellite WFV4 image data are obtained from the official website of the China Resources Satellite Application Center (http://www.cresda.com/ (accessed on 18 March 2022)). The 30-m NASADEM data come from the National Aeronautics and Space Administration (NASA) (https://earthdata.nasa.gov/esds/competitive-programs/measures/nasadem (accessed on 18 March 2022)). GlobeLand30 data come from the website (http://globallandcover.com/ (accessed on18 March 2022)).

**Conflicts of Interest:** The authors declare no conflict of interest.

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
