# Peer review of "Satellite-Based Optimization and Planning of Urban Ventilation Corridors for a Healthy Microclimate Environment"

_sustainability, doi:10.3390/su152115653_

Round 1

Reviewer 1 Report

Comments and Suggestions for Authors

This manuscript (sustainability-2617154) tries to analyse the urban ventilation corridor construction methods based on the local circulation patterns. The thermal field and PM 2.5 concentration field in Shanghai, China were retrieved based on remote sensing technique. In addition, a Weather Research and Forecasting (WRF) model coupled with a multilayer urban scheme Building Effect Parameterization (BEP) model is used to numerically simulate and analyse the wind field in the study area. The study fits the aim and scope of this journal and the amount of the work is enough. Overall, this is a useful manuscript. Some detailed suggestions and comments are presented as follows:

- 1. Some quantitative results should be included in the Abstract. The current abstract is a bit qualitative and lacks some important result descriptions.

- 2. From Line 85 to 87, the authors have mentioned that: "The above researches on urban ventilation corridors mainly used RS and GIS techniques to analyze their spatial compositions and action mechanisms, however, lacked analyses of the urban wind environment and the spatial distribution of air pollution". I suggest these statements should be explained more convincingly.

- 3. From Line 54 to 64, a number of previous related studies have been mentioned, and I suggest the authors to summarize the limitations and shortcomings of these studies (shown below for examples).

A New method of simulating urban ventilation corridors using circuit theory. Sustainable Cities and Society, 2020, 59, 102162.

Performance evaluation on multi-scenario urban ventilation corridors based on least cost path. Journal of Urban Management, 2020, 10 (1), 3-15.

- 4. In Section 2. Description of Study Area and Data Used, the conditions of urban heat island and air pollution in Shanghai should also be introduced briefly.

- 5. Figure 1. Map of the study area. (a) Location of Shanghai in the Yangtze River Delta. (b) Districts in 126 Shanghai: please provide the coordinates of this study area.

- 6. I suggest all the datasets used in this study should be summarized in a new Table, including the years, spatial resolution, data sources, and website links.

- 7. The data should be displayed in GIS spatial maps if possible.

- 8. In Section 3.2.1. Analysis of the thermal environment, more detailed information about the calculation of the intensity of UHI effect, such as the equations, should be explained clearly. Please refer to the below reference.

Measuring the relationship between morphological spatial pattern of green space and urban heat island using machine learning methods. Building and Environment, 2023, 228, 109910

- 9. Please provide a technical roadmap of this research using a new figure.

- 10. I suggest to condense the contents of the Implementation and Results sections as the information is quite dense. In this respect, the document reads more like a dissertation or thesis. Simplify and make the information more concise.

- 11. In the Discussion and Conclusion Section, please provide and discuss the major shortages of this current research.

- 12. Takeaway for practice is also encouraged to be included in this manuscript. It would be better to present your implications and recommendations for both local and international practice.

Comments on the Quality of English Language

Some minor editing of English language required.

Reviewer 2 Report

Comments and Suggestions for Authors

Manuscript ID: sustainability- 2617154_review

Title:  Satellite-based Optimization and Planning of Urban Ventila-2 tion Corridors for a Healthy Microclimate Environment 3

Deming Gong 1, Xiaoyan Dai 1,*, Liguo Zhou 1,**

Journal: Sustainability

The article addresses a topic of interest and relevance, but presents some minor methodological problems. I have read the study thoroughly. I recommend minor revisions for this study based on my comments below.

Ø  The abstract is too generic. The abstract should consist of an easy to understand summary of the study, its methodology and obtained results. It is difficult to understand the research questions and what is expected from the study. How do these studies set up the context for this? Given the number of studies in this research area; what gap will this study fill in the literature? These results are very general.

* All figures have low visual quality and resolution.

* Is the survey conducted only in March sufficient? Does it represent the whole year? After all,

* UHII what does it mean? What is the expandation of this. There are a few more like this "UHII”

Ø  Materials and Methods are  descriptive and provide any information related to the input data and the methodology used in the study. Method description is sufficient. However, it would be better if a workflow diagram is provided.

Ø  The discussion section is missing. Regarding the discussion, the authors should further compare their findings with references. As you included a very wide range of background information, a more structured illustration of these background literature references could be promoted to add another benefit to the paper. Link your findings with those from previous studies and this will also help make more broader conclusions. Discussion- Compare your own results in the discussion. It is not correct to give only literature information. This part is very important. It would be good to refer to the following publications while preparing the discussion section. In conclusion, you should only provide main outcomes of the study and discuss limitations of the study and recommendation for future researches.

https://doi.org/10.1016/j.uclim.2021.101052
https://doi.org/10.1007/s41748-023-00340-6

https://doi.org/10.1016/j.buildenv.2022.109210
https://doi.org/10.1007/s11356-023-28553-2

https://doi.org/10.1016/j.envres.2023.116887

Ø  A conclusion should be made based on the findings obtained, and it is essential to provide an interpretation of the results. What conclusions have been drawn for sustainable urbanization? How will these be translated into planning? Concrete takeaways should be included

Ø  The present work could be interesting for the future urban planning for suistanable cities. But this research has minor methodological flaws. Discussion should be added because it has not been done. Please, relevant literatures should be consulted and discussed. Results should also be explained in clear terms.  How will results affect urban design and planning? At this point, the paper will need to have revisions restructuring. At this point, the paper will need to have minor revisions restructuring.

Reviewer 3 Report

Comments and Suggestions for Authors

The research is relatively novel and has certain reference value for urban planning and design.  I have the following questions,

1. An introduction to the progress of urbanization in the study area in recent decades can be added in the study area introduction in lines 111 to 124 in the part of Description of Study Area and Data Used.

2. In data sources part, line 113-148, the Landsat sensor data for calculating LST selected 10:25 on August 16, 2020, while the data for calculating PM2.5 selected 10:52 on February 21, 2021. Why not choose the same time period? Please explain.

3. I did not see the validation of WRF simulation results for this study, is it not necessary?

4. Figure 10 shows R=0.85, please confirm whether it is R2?

Comments on the Quality of English Language

Can be further improved

Round 2

Reviewer 1 Report

Comments and Suggestions for Authors

Although this revised version has been improved to some degree, there are still several major issues in it:

- 1. The analysis of the response of urban ventilation corridors lacks depth and only stays at the level of data analysis.

- 2. In the Results Section, how to link the findings and conclusions in this paper with the previous findings and conclusions?

- 3. The results and discussions, in particular, failed to highlight the value of using this approach compared to traditional conclusions. There are few quantified comparisons that show differences or improvements over the more common research findings.

- 4. In the case that urban heat island effect will be related to meteorological factors such as humidity, rainfall, and wind speed/wind direction, it seems that this study did not consider these factors.

- 5. The conclusion section also needs to be further deepened after the follow-up improvement, rather than just some simple and repeated data description.

- 6. Please provide the English editing service certificates.

Comments on the Quality of English Language

Please provide the English editing service certificates.

Round 3

Reviewer 1 Report

Comments and Suggestions for Authors

I have no further comments.

Comments on the Quality of English Language

Minor editing of English language required.